# Prompt-guided and multimodal landscape scenicness assessments with vision-language models

Alex Levering[1,2], Diego Marcos[3], Nathan Jacobs[4], Devis Tuia[5]*

1 Laboratory of Geo-Information Science and Remote Sensing, Wageningen University, Wageningen, the Netherlands, 2 Instituut voor Milieuvraagstukken, Vrije Universiteit Amsterdam, Amsterdam, the Netherlands, 3 Inria, Université de Montpellier, Montpellier, France, 4 McKelvey School of Engineering, Washington University in St. Louis, St. Louis, MO, United States of America, 5 Ecole Polytechnique Fédérale de Lausanne, Environmental Computational Science and Earth Observation Laboratory, Sion, Switzerland

* devis.tuia@epfl.ch

## Abstract

Recent advances in deep learning and Vision-Language Models (VLM) have enabled efficient transfer to downstream tasks even when limited labelled training data is available, as well as for text to be directly compared to image content. These properties of VLMs enable new opportunities for the annotation and analysis of images. We test the potential of VLMs for landscape scenicness prediction, i.e., the aesthetic quality of a landscape, using zero- and few-shot methods. We experiment with few-shot learning by fine-tuning a single linear layer on a pre-trained VLM representation. We find that a model fitted to just a few hundred samples performs favourably compared to a model trained on hundreds of thousands of examples in a fully supervised way. We also explore the zero-shot prediction potential of contrastive prompting using positive and negative landscape aesthetic concepts. Our results show that this method outperforms a linear probe with few-shot learning when using a small number of samples to tune the prompt configuration. We introduce Landscape Prompt Ensembling (LPE), which is an annotation method for acquiring landscape scenicness ratings through rated text descriptions without needing an image dataset during annotation. We demonstrate that LPE can provide landscape scenicness assessments that are concordant with a dataset of image ratings. The success of zero- and few-shot methods combined with their ability to use text-based annotations highlights the potential for VLMs to provide efficient landscape scenicness assessments with greater flexibility.

## Introduction

In these times where urban expansion is prevalent, maintaining the quality of our landscapes is increasingly important, as it plays a significant role in our overall well-being. Beyond their visual appeal, scenic landscapes offer a multitude of benefits, both tangible and intangible. They provide us with a sense of tranquility, an escape from the stress of our daily lives, and a connection to the natural world. Moreover, research has shown that exposure to scenic

**Data Availability Statement:** Our data is openly available in the following Zenodo repository: https://zenodo.org/records/12653736 - it contains the prompts gathered in this research, and the embeddings extracted using both models. We also

provide reproduction scripts and scripts to download the original images on the following GitHub page: https://github.com/ahlevering/scenicness_prompting.

**Funding:** The author(s) received no specific funding for this work.

**Competing interests:** The authors have declared that no competing interests exist.

environments is associated with many positive effects. Exposure to natural environments is shown to be beneficial to our attention span [1, 2], our stress management [1, 3], and our overall happiness [4]. Scenic landscapes also instill a sense of comfort, tranquility, and safety [5]. Beyond personal health benefits, scenic landscapes are a driver for tourism [6], as well as cultural ecosystem services [7, 8].

One way to protect natural environments is to document their presence and evaluate their aesthetic appreciation by humans, or *scenicness*. Improvements in computer vision methods in the past decade have made it possible to predict scenicness directly from images. Meanwhile, increased internet connectivity across the globe has enabled crowdsourcing at unprecedented scales. These developments combined have resulted in the exploration of landscape aesthetic preferences directly from images in a data-driven setting [9–11]. However, such research efforts in turn suffer from the large amounts of annotated images needed to train deep-learning models. As a result, only a few studies have attempted to study aesthetic preferences on a local scale using deep learning and data-driven methods.

Recent research developments have seen the convergence of natural language processing and computer vision into *Vision-Language Models* (VLM), such as the CLIP model [12]. Trained on images gathered from the internet with their corresponding text captions, these models are able to relate the content of images to textual descriptions to determine their relatedness. CLIP and similar models have since closed the gap between data-efficient learning strategies such as few-shot learning and fully supervised training on large datasets. Rather than needing tens of thousands of annotated examples, VLMs are able to use a fraction of data and still perform competitively on many tasks [12–14]. Therefore, VLMs hold the potential to enable accurate data-driven analyses of many tasks using small-scale datasets.

In this study, we evaluate the effectiveness of VLMs for quantifying and mapping landscape scenicness at scale, using both rated image datasets and a new text-based annotation approach. Firstly, we explore the potential of VLMs for landscape scenicness assessments using data-efficient learning regimes using a dataset of rated images. We explore a few-shot prediction setting with linear probes and a zero-shot setting using prompts of opposing landscape scenicness concepts. Secondly, we introduce a new annotation method that leverages the ability of VLMs to associate text with images, which we refer to as *Landscape Prompt Ensembling* (LPE). We demonstrate that ensembles of rated text descriptions provided by volunteers can provide landscape scenicness assessments without the need for an image dataset while annotating.

## Scenicness prediction with machine learning

The prediction of landscape aesthetic qualities from images became possible with the rapid improvements of machine- and deep learning models in the last decade. Dubey et al. [10] introduced the Place Pulse 2.0 dataset, where labelers were shown two images of urban streetscapes and asked to vote on their preferred image with respect to six different qualities. Among the adjectives that volunteers were asked to rate was *beautiful*. Subsequent research has explored a variety of topics, such as relating adjective predictions to objects in urban environments to determine their influence [15], determining their spatial patterns through land use classes [16], and using them to synthesise ideal neighbourhoods [17]. In further research concerning the aesthetic quality of urban spaces, Verma et al. [18] crowdsourced scenicness ratings on a local scale and used these ratings to measure the effects of changing conditions on scenicness. Christman et al. [19] relate objects that are implicitly assumed to be scenic or unsightly (e.g., flowers or trash bags) to the walkability of neighborhoods. Chen and Biljecki [20] predict the design and aesthetic quality of urban areas in Singapore and compare their importance to features of the landscape.

Directly related to our research is the research performed on the estimation of scenicness as a quantitative score. The dataset used for this purpose is the ScenicOrNot (SON) dataset [9], a crowdsourcing effort to rate the scenicness of images across the entirety of Great Britain and the Isle of Man. Early works trained convolutional neural networks to regress the scenicness score directly [21]. Subsequent research has largely focused on understanding how scenicness relates to its environment. Havinga et al. [8] first extracted landscape features such as their scene class, then related them to scenicness through a Random Forest approach. Further research has attempted to use CNNs for explicit relations between intermediate concepts such as scenes or land cover to relate them to scenicness in an interpretable manner [22, 23]. Finally, Arendsen et al. [24] attempted to discover how concepts that have not been trained on relate to scenicness. SON has also been used to study scenicness from new perspectives, such as a hybrid ground-and-overhead imagery perspective [25], as well as through satellite imagery [26].

## Vision-language models

While much attention has been devoted to classification tasks using VLMs and data-efficient learning methods [12, 27, 28], comparatively little has been done to develop methods compatible with regression tasks. Li et al. introduced OrdinalCLIP, which uses an ordinal output space in order to utilise the classification-based few-shot methods [29]. Hentschel et al. [30] trained a linear probe on CLIP image features on the regression task of image photographic aesthetics understanding in a few-shot setting, with competitive results compared to a fully-trained baseline. Ke et al. [31] introduced VILA, a VLM fine-tuning and zero-shot prediction pipeline for image aesthetics. The authors fine-tuned a pre-trained VLM on image-text captions, where the text captions provide feedback on the given image about its aesthetic properties or qualities. The authors then test the zero-shot regression performance of their model by using the contrast between two prompts (e.g., "good photo" versus "bad photo"), where the model's confidence in the positive prompt is assumed to be correlated with photographic aesthetic beauty. Their method delivered competitive results compared to fully-trained baselines. The authors also experiment with other tasks, such as photo-aesthetic captioning. Recent further developments on VLMs have resulted in the emergence of Multimodal Large Language Models (MLLMs) for downstream tasks, such as visual-question answering without fine-tuning. Liang et al. [32] investigated the use of MLLMs to query city changes on a variety of tasks, such as green space and street facade improvements, finding good correspondence with collected reference opinions.

## Data and methods

### Data

**ScenicOrNot.**   For the prediction of landscape scenicness, we require a reference dataset with image ratings. For this purpose, we use the ScenicOrNot (SON) dataset [9] as our reference dataset. SON consists of a collection of approximately 217000 images with ratings provided by anonymous volunteers on a scale between 1 (most unsightly) and 10 (most beautiful). The images used by SON are obtained from the Geograph UK project (https://www.geograph.org.uk/). The images are stored with geolocation information and can therefore be used for mapping. To acquire scenicness ratings for the images, the authors then used a crowdsourcing website, wRhich loads a random image and asks the visitor to rate it. Images were only included in the dataset if they had at least 3 ratings. The SON website provides a file that contains the ratings of each image, along with the image path on the Geograph website. After removing images that had been taken offline from the ratings file, we are able to use a total of

212 104 images. We show the spatial distribution of the rated images and corresponding scores in Fig 1.

**Landscape prompt annotations.** For the SON dataset, volunteers were asked to provide ratings of images directly, which creates a dataset with image ratings. Such an approach provides accurate labels for individual images, but implicitly, such ratings are not informative about individual voter aesthetic preferences for certain landscape types. Instead, we propose to annotate a dataset that captures the explicit landscape preferences of each volunteer by leveraging the ability of VLMs to relate text descriptions to image content. We refer to our annotation method as *Landscape Prompt Ensembling* (LPE). To create a prototype dataset, we involved a group of non-expert volunteers and asked them to provide prompt-rating pairs that describe the landscapes of the United Kingdom through an anonymous survey. In this survey, we showed four example images of scenes with descriptions and ratings that voters could imagine. We then asked them to imagine and write out their own landscape impressions of the United

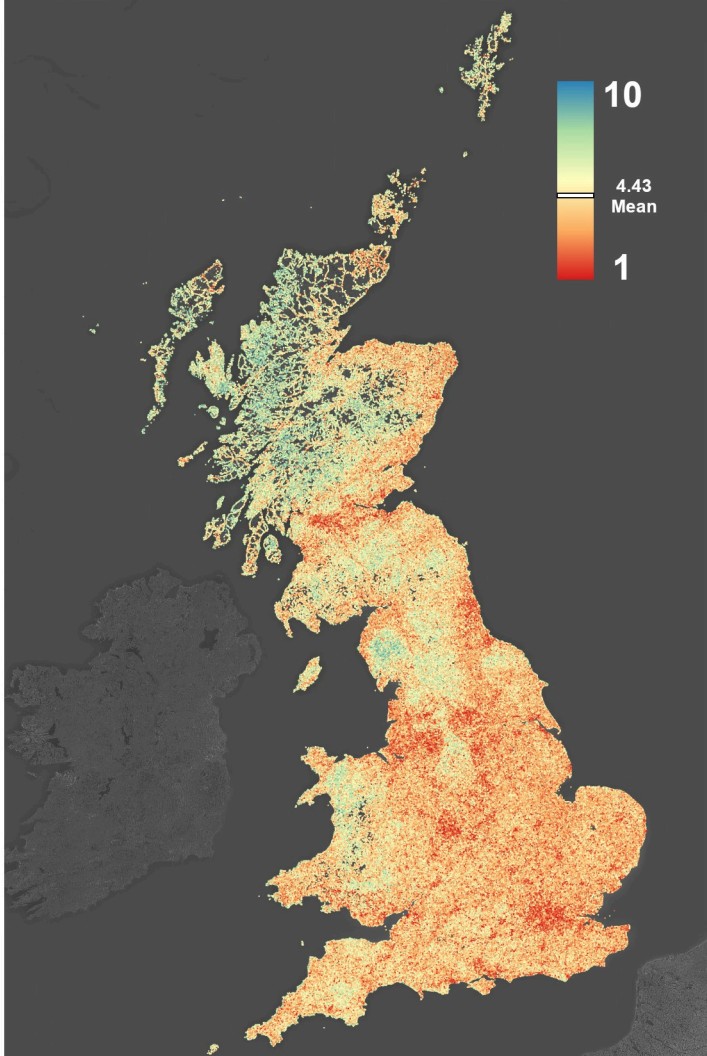

**Fig 1. ScenicOrNot image ratings plotted at their georeferenced coordinates for the entirety of Great Britain and the Isle of Man.** Values range from 1 to 10, where 10 is the most scenic, with an average scenicness rating of 4.43.

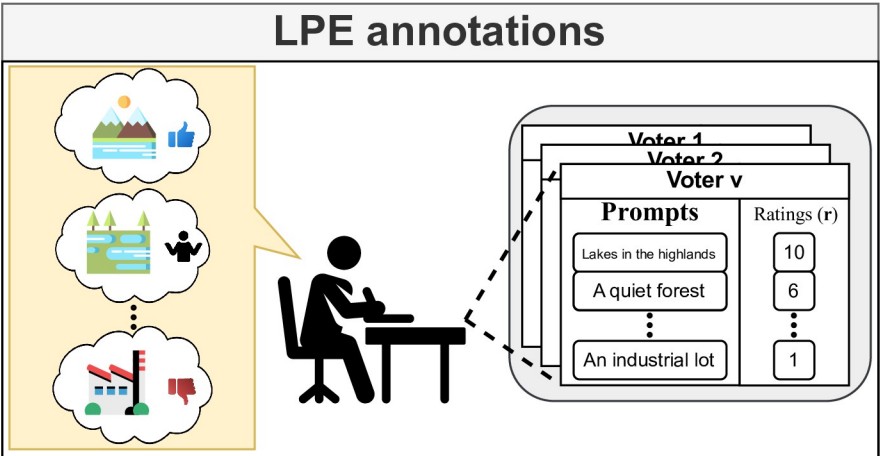

**Fig 2. Rated landscape prompt annotation workflow.** Voters are asked to imagine landscapes that they like or dislike. They are then asked to write a description of these landscapes and to give it a rating between 1 and 10. The resulting dataset is a collection of landscape descriptions and their associated ratings from every voter.

Kingdom. As such, in our annotation process, we do not need an extensive image dataset during annotation, instead relying on the volunteers' imagination. We also surveyed the confidence that voters have in their ratings and whether or not they have visited the United Kingdom before. In total, we received 45 responses. We removed empty responses and responses that did not provide prompts in the requested format. Of these 27 remaining responses, the median number of prompts provided was 4. In total, voters provided 136 prompts. 18 voters provided more than 1 prompt at once. Of the 19 respondents for which confidence information is available, the average voter confidence was 3.36 out of 5, and 57% of the labelers had visited the United Kingdom before. We give an overview of our annotation process in Fig 2. We show how our LPE annotations can be used to generate image ratings, such as in the SON dataset, through early ensembling and late ensembling.

## Methods

In our research we explore how well VLMs can be used in efficient data regimes for the task of scenicness prediction. For this purpose, we use recent advances in model pre-training and we test the utility of the text encoder of our VLM. After testing the robustness of VLMs features in a few-shot setting, we propose a method for zero-shot prediction based on contrastive prompting. We then demonstrate the power of VLMs in assessing landscape scenicness through text descriptions by using Lnadscape Prompt Ensembling (LPE), where we generate image ratings from prompt-based landscape annotations.

 **CLIP.** The VLM feature extractor we use in our experiments is a CLIP-pre-trained transformer [12]. This model consists of an image feature extractor and a text feature extractor, which have been jointly optimised during pre-training so that they share the same embedding space. A simplified example of this model is given in Fig 3. The image encoder encodes each image to a vector $\mathbf{x}$, and the text encoder encodes each textual prompt to a vector $\mathbf{t}$. The vision/text embedding space learned by CLIP is multimodal and aligned, in the sense that a prompt and a corresponding image would be mapped in the same location of the embedding space. We use the image and text encoders as provided in the checkpoints of HuggingFace (details in Section "Experimental set-up").

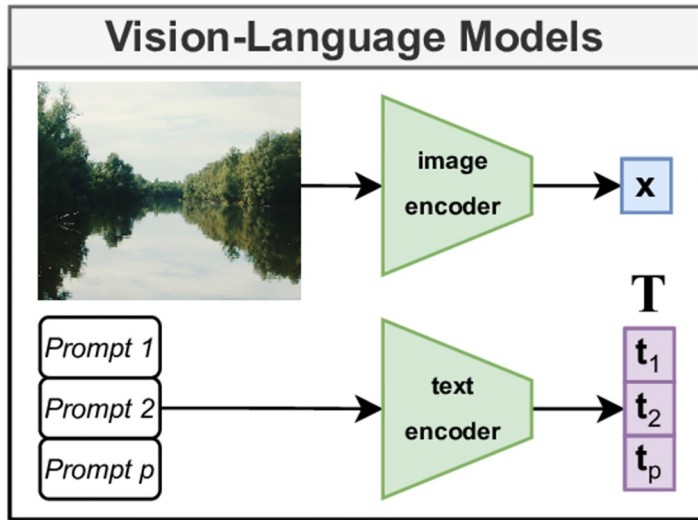

**Fig 3. Simplified overview of VLM models.** VLMs use two separate encoders, which map images and text to the same latent space.

**Few-shot learning of VLM features.** In this setting, we study how well VLMs can predict scenicness using only the image encoder through few-shot learning. In other words, we only consider the image feature vector **x** extracted by the image feature extractor without using any text features. We optimise a single linear layer, which maps each image feature vector to a scenicness score prediction $\hat{s}$:

$$\hat{s} = \mathbf{x}^\mathsf{T}\mathbf{w} + b, \tag{1}$$

where **w** is a vector of weights and $b$ is a bias term. The linear layer is then optimized using a squared error loss:

$$L_{scenic} = (s - \hat{s})^2, \tag{2}$$

where $s$ is the SON rating matching the input image.

**Contrastive prompting.** In our second experiment we test the zero-shot capabilities of the text encoder. In this setting we explore which prompt formulations are suitable for predicting landscape scenicness. Our method uses a shared prompt context with a pair of antonyms to derive the model's preference for the positive concept of the antonym pair. The prompt context is a general sentence such as "A photo of a landscape that is [. . .]", where the text in brackets is replaced by either synonyms or antonyms of scenicness. The use of this prompt is intended to provide good discriminative text features for the task, and its importance was first demonstrated by Zhou et al. [13]. For a given set of two prompts comprised of one scenicness synonym and one scenicness antonym, we first calculate the text feature activation matrix **T** of size $(d_t \times 2)$, where $d_t$ is the number of features of the text encoder of the VLM. For a given image activation vector **x** of size $d_i = d_t$, we can then calculate the logits and use a Softmax activation function to determine the activation of the prompts for the image under consideration:

$$\mathbf{a} = \text{Softmax}((\mathbf{x}^\mathsf{T}\mathbf{T}) * Z), \tag{3}$$

where the scaling factor $Z$ has been estimated empirically during the pre-training of CLIP. Note that the activations of **a** sum to 1. We can assume that the model's confidence in the

**Fig 4. Prediction pipeline for the contrastive prompting method.** We first define a positive and a negative prompt with a shared prompt context. Then, we use the model's confidence in the positive prompt and rescale it between 1 and 10 as the scenicness prediction $\hat{s}$ for a given image.

positive scenicness synonym prompt at index 1 is linearly related to the scenicness of a given image. We rescale the model's confidence in the positive prompt at index 1 to the 1 to 10 range of the scenicness reference data to derive the predicted scenicness score $\hat{s}$:

$$\hat{s} = (a_1 * 9) + 1 \tag{4}$$

The resulting predictions can then be compared to the reference scenicness scores to determine the performance of the given prompt construction. As this is a zero-shot method, no parameter updates are performed. We give a graphical overview of our approach in Fig 4.

**Landscape prompt ensembles.**   In this experiment, we test two ways of deriving image scenicness ratings from the rated text prompt annotations described in the "Landscape prompt annotations" section. In this annotation format, each voter $v \in V$ provides a list of landscape prompts with matching ratings. For each voter $v$, we can extract a matrix of text features $\mathbf{T}_v$ from the prompts with a VLM. For each voter, we also have a vector of ratings, $\mathbf{r}_v$. We test how well our prompt-based annotations can be used to generate image ratings through two types of ensembling methods, *late ensembling* and *early ensembling*, which calculate the prompt activations in differing ways.

*Early ensembling*. With early ensembling, we hypothesise that if many voters provide many prompts, then there will be a few prompts that most accurately describe each image. For instance, a photo that depicts a river in a forest benefits from having the river mentioned, as it provides a more detailed description than just a prompt about forests in general. For this purpose, we consider all prompts provided by all voters at once. First, we concatenate the encoded text feature matrices of the prompts of all voters into a single text feature matrix $\mathbf{T}$:

$$\mathbf{T} = \begin{bmatrix} \mathbf{T}_1 & \mathbf{T}_2 & \dots & \mathbf{T}_V \end{bmatrix}. \tag{5}$$

We can then calculate the activations of each prompt with Eq (3) to derive the prompt activation vector $\mathbf{a}$. In this setting, $\mathbf{a}$ contains the activations of all prompts provided by all voters for the given image through a single Softmax. As such, it represents the probability that each prompt best matches the image. We then concatenate the ratings of all voters into a single ratings vector $\mathbf{r}$ such that it matches the shape of the activation vector $\mathbf{a}$:

$$\mathbf{r} = \begin{bmatrix} \mathbf{r}_1 & \mathbf{r}_2 & \dots & \mathbf{r}_V \end{bmatrix}. \tag{6}$$

By multiplying the probability that each prompt best matches the given image with its provided scenicness rating, we can then compute the predicted scenicness score of the image from

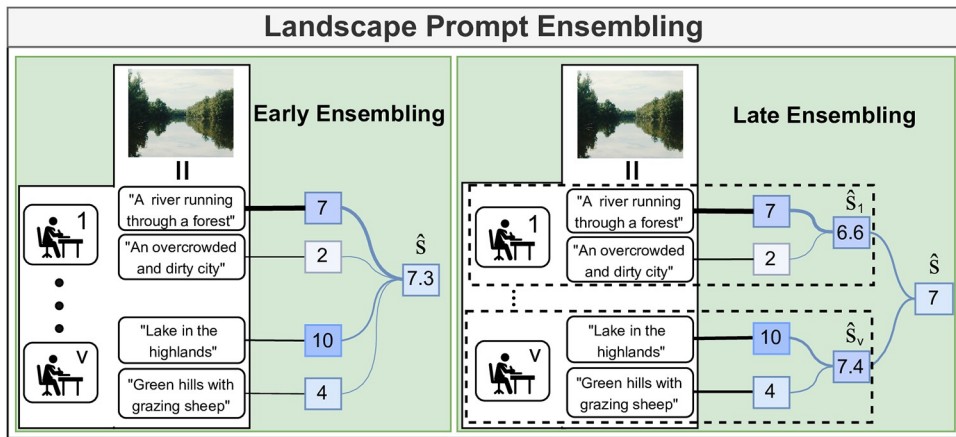

**Fig 5. Methods of ensembling for generating image ratings from landscape prompts.** In early ensembling, we use the likelihood that any given prompt matches an image. The likelihood of each prompt is then multiplied by its voter-provided scenicness rating to determine the scenicness score of a given image. In late ensembling, we consider this weighted likelihood for the prompts of each voter individually to calculate a voter-specific scenicness score. We then average across all voter scenicness scores to calculate the scenicness score for the image.

all contributions:

$$\hat{s} = \mathbf{a}^{\mathsf{T}}\mathbf{r}. \tag{7}$$

*Late ensembling.* In the late ensembling case, we hypothesise that having many voters with less detailed prompts will capture the variance of landscape preferences on a macro-scale, similar to the variance observed for the ratings of individual images in SON. For each voter $v \in V$, we first extract the activations of their provided prompts:

$$\mathbf{a}_v = \text{Softmax}((\mathbf{x}^{\mathsf{T}}\mathbf{T}_v) * Z). \tag{8}$$

To calculate the scenicness score for a given image, we can then multiply the activation vector of each voter with the ratings of the voter and take the average across the total number of voters to calculate the image scenicness score:

$$\hat{s} = \frac{\sum_{v=1}^{V} \mathbf{a}_v^{\mathsf{T}}\mathbf{r}_v}{V} \tag{9}$$

Both prompt ensembling methods are graphically displayed in Fig 5.

## Experimental set-up

For our VLM, we use the ViT-L/14 variant of the CLIP pre-trained models [12]. We freeze the feature extractors and do not optimise them during any of our experiments. We evaluate our methods using the root mean squared error (RMSE), the coefficient of determinant $R^2$ and Kendall's $\tau$, which is a ranking coefficient ranging from $-1$ (all values are inversely ranked) to 1 (all values are ranked perfectly in order) [33]. Experiments with other VLM pre-training methods such as SigLIP [34] did not show significant improvements. Results for SigLIP embeddings can be found in S1 Appendix. We release the code for our experiments on GitHub (https://github.com/ahlevering/scenicness_prompting), and our data is available on Zenodo (https://doi.org/10.5281/zenodo.12653736).

**Few-shot learning.**   We run experiments with a total number of samples of $n \in (25, 50, 100, 250, 500)$. We cluster all landscape photos in SON, run k-Means (k = 25), and pick $n/k$ samples of each cluster. We optimise the linear layer using stochastic gradient descent. We use 5-fold cross-validation over the learning rates 1e-2 through 1e-8 with increments of 0.1, and we use the model with the best training $R^2$ during testing, as the $R^2$ is more stable than the RMSE on this task. We compare the performance of probes trained on embeddings from the ViT-L/14 model to an ImageNet-pretrained ConvNeXt-Large model [35] to determine the effect of web-scale pre-training for linear probes. We use the same training regime for this model. Lastly, we compare the performance of both few-shot trained models to a CLIP pre-trained ViT-L/14 model fine-tuned on the entire dataset. We train this baseline model with a learning rate of 1e-3 with the AdamW optimizer [36]. We randomly sample 10% of the dataset for testing. Of the remaining 90% of the dataset, we use 85% for training and 15% for validation.

**Contrastive prompting.**   We test different prompt configurations in our contrastive prompting experiments. We consider six prompt contexts: *"A photo that is"*, *"A photo that is extremely"*, *"A photo of an area that is"*, *"A photo of an area that is extremely"*, *"A photo of a landscape that is"*, *"A photo of a landscape that is extremely"*. We test these prompt contexts to evaluate two aspects, namely 1) the importance of emphasis on landscapes (*"area"* and *"landscape"*), and 2) the effect of adding a superlative (*"extremely"*). For the positive and negative concepts, we use the synonyms and antonyms of "scenic" as listed on a thesaurus website (https://www.thesaurus.com). The synonyms that we use are *"breathtaking"*, *"grand"*, *"spectacular"*, *"striking"*, *"dramatic"*, *"panoramic"*, *"impressive"*, *"beautiful"*, and *"picturesque"*. For the antonyms, we use *"normal"*, *"usual"*, *""dreary"*, *""ugly"*, *""ordinary"*, *"despicable"*, and *"gloomy"*. We test all possible combinations of context, positive, and negative concept choices, which results in 378 unique prompt configurations. To determine the stability of prompt configurations we compute the metrics on the full dataset as well as the samples used in the 25-sample few-shot learning case. In doing so, we test if a single representative sample from each type of landscape in the SON dataset will result in similar highly-performing prompt configurations compared to the full dataset.

**Landscape prompt ensembles.**   Both the early and late ensembling methods do not require parameterization and can be run as-is. We instead analyse the trade-off between the number of voters and the number of prompts they provide in the case of late ensembling. We test the effect of using ensembles with a minimum of 2, 5, 8, or 10 prompts. In doing so, we can test if it is better to have many voters with a small number of prompt suggestions or to have a few voters who provide more accurate prompt suggestions.

## Results and discussion

### Few-shot experiments

Fig 6 illustrates the performance of the few-shot linear probes. Our results suggest that it is possible to accurately predict scenicness with far fewer labelled examples than has previously been attempted. Compared to the fully-trained baseline model, which was pre-trained on ImageNet, we find that the linear probe based on the ViT-14/L CLIP pre-trained transformer is nearly as accurate when using $n = 500$ labelled samples. However, despite requiring approximately 342 times fewer training samples than the fully-trained baseline, this can still be considered a non-trivial labelling effort as all of the images in SON are rated by at least 3 voters. Reducing $n$ further, we observe that a model that uses just $n = 100$ samples has approximately 10% lower accuracy but uses 5 times fewer labels. Our findings suggest that a few-shot linear probe can provide an adequate estimation of landscape scenicness even with a limited number of samples.

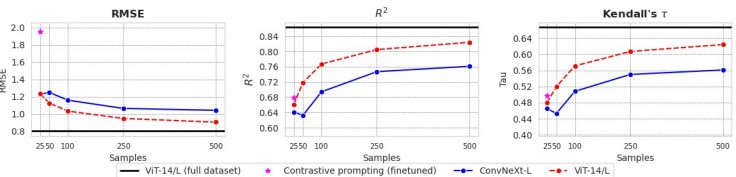

**Fig 6. Results for the few-and zero-shot prediction methods.** The black line shows the performance of a ViT-14/L model initialised with CLIP pre-trained weights trained on the complete SON dataset. The line represents the same model, where only the linear probe is fine-tuned in a few-shot setting. The red line shows the performance of the ViT-14/L vision transformer pre-trained using CLIP. While both the ConvNeXt-Large and ViT-14/L models provide adequate few-shot performance, the transformer model with web-scale pre-training and more parameters performs substantially better. When using 25 samples to estimate the best prompt combination, our zero-shot contrastive prompting method (shown in magenta) shows superior ranking performance compared to the few-shot models, although predictions are further off from the reference values as evidenced by the high RMSE.

## Contrastive prompting

In Fig 7, we show the performance distribution for each choice of prompt context. From the distributions over the full dataset, we observe that the method is sensitive to the exact formulation of the prompt and that both the context and the choice of synonyms and antonyms are of importance. The best-performing prompt context on our dataset (*"A photo of an area that is extremely"*) still has outliers, which produce a poor fit. In the best-case scenario, the contrastive prompting method outperforms the few-shot probe on the task of ranking samples when only a limited number of samples are available to train the probe.

In the pure zero-shot setting, it is not possible to know a priori which prompt combination is optimal. In Fig 7, we therefore also show the performance of each prompt context when applied to the samples of the $n = 25$ case of the few-shot setting, which we refer to as the *calibration set*. The resulting distributions for each prompt context are highly similar, which suggests that a few labelled samples may be used to tune the prompt configuration. We show the metric performance of the best prompts on the fine-tuning set in Table 1. We take the best-performing prompt configuration and apply it to the full dataset. Fig 6 highlights that it outperforms a linear probe on the task of ranking samples, but it also has a very high RMSE and a marginally worse $R^2$. These results suggest that the optimal prompt configuration for this task is maximally discriminating between the positive and negative concepts.

## Prompt ensembles

In this section, we compare the computed image scenicness ratings of our LPE ensembling methods to the SON image ratings. In Table 2, we show the numerical metrics of both of our

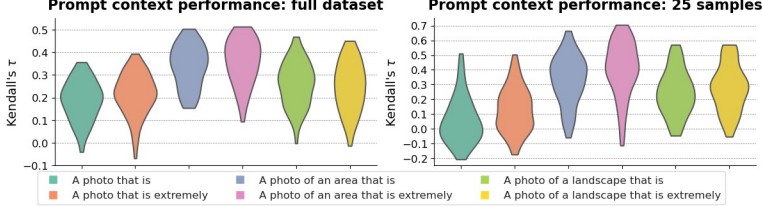

**Fig 7. Distribution of Kendall's $\tau$ of each of the six prompt contexts of the contrastive prompting method when evaluated on the entire dataset (left) and when applied to only the $n = 25$ samples from the few-shot learning case (right).**

**Table 1. Metric performance of the top-five contrastive prompting configurations evaluated on the calibration set.** The highly-performing prompts in this setting are similar to those observed for the full dataset.

| Context | Positive concept | Negative concept | RMSE | $R^2$ | $\tau$ |
|---|---|---|---|---|---|
| area that is extremely | breathtaking | usual | **1.52** | **0.841** | **0.705** |
| area that is | panoramic | normal | 2.31 | 0.801 | 0.664 |
| area that is extremely | panoramic | ugly | 2.64 | 0.813 | 0.658 |
| area that is extremely | breathtaking | normal | 3.43 | 0.744 | 0.644 |
| area that is extremely | panoramic | usual | 3.50 | 0.729 | 0.644 |

**Table 2. Comparison of our LPE method with SON image scores.** Late fusion results in image scenicness ratings that are closer to the SON image ratings, and including more voters results in a higher degree of agreement with SON, even if these voters provide fewer prompts per person.

| Method | voters | Total Prompts | RMSE | $R^2$ | $\tau$ |
|---|---|---|---|---|---|
| Early | 27 | 137 | 3.32 | 0.544 | 0.390 |
| **Late** | | | | | |
| >=2 prompts | 18 | 129 | 2.60 | **0.692** | **0.485** |
| >=5 prompts | 10 | 105 | **2.10** | 0.654 | 0.447 |
| >=8 prompts | 5 | 74 | 2.72 | 0.673 | 0.479 |
| >=10 prompts | 3 | 49 | 3.32 | 0.623 | 0.45 |

prompt ensembling methods when compared to SON labels. The results demonstrate that late prompt ensembling is a more effective ensembling method than early prompt ensembling by a substantial margin. We hypothesise that this is the result of the implicit variance in the appreciation of landscapes between voters, which is not accounted for during early ensembling. Even if the VLM model retrieves the most accurate prompt for a specific image with high confidence, the rating assigned to this image could be strongly influenced by the voter's personal preferences. Therefore, late ensembling appears to be the more reliable method for generating image ratings from the rated prompt annotations. The numerical comparison with SON also suggests that it is better to have many voters provide prompts as opposed to a small subset of voters who provide many prompts. Interestingly, the quality or diversity of each ensemble appears to matter less than the number of individual ensembles. Therefore, a higher number of respondents is more important than ensuring that all respondents provide many prompts at once. Using our small dataset, we even find that late prompt ensembling can be more effective at acquiring scenicness rankings of images than fine-tuning a linear probe using 25 or fewer image labels. Our results demonstrate that LPE shows potential for landscape scenicness assessments without the need for an image dataset during annotation.

## Geographical prediction patterns

In Fig 8, we show the spatial prediction results for all methods used in our study. We observe a few notable differences between methods. In the few-shot learning setting, the main difference between using $n = 25$ samples (1 example per cluster centroid) and $n = 500$ samples (20 examples per cluster centroid) is an increase in the model's ability to predict values at both ends of the distribution, such as the very low scenicness in cities and the very high scenicness of the Scottish highlands. When considering the zero-shot contrastive prompting approach, we observe that the most effective prompt combinations are all highly discriminative between the scenicness synonyms and antonyms. The resulting predictions do follow the distribution of scenicness of the reference data, but with hardly any predictions in the middle of the

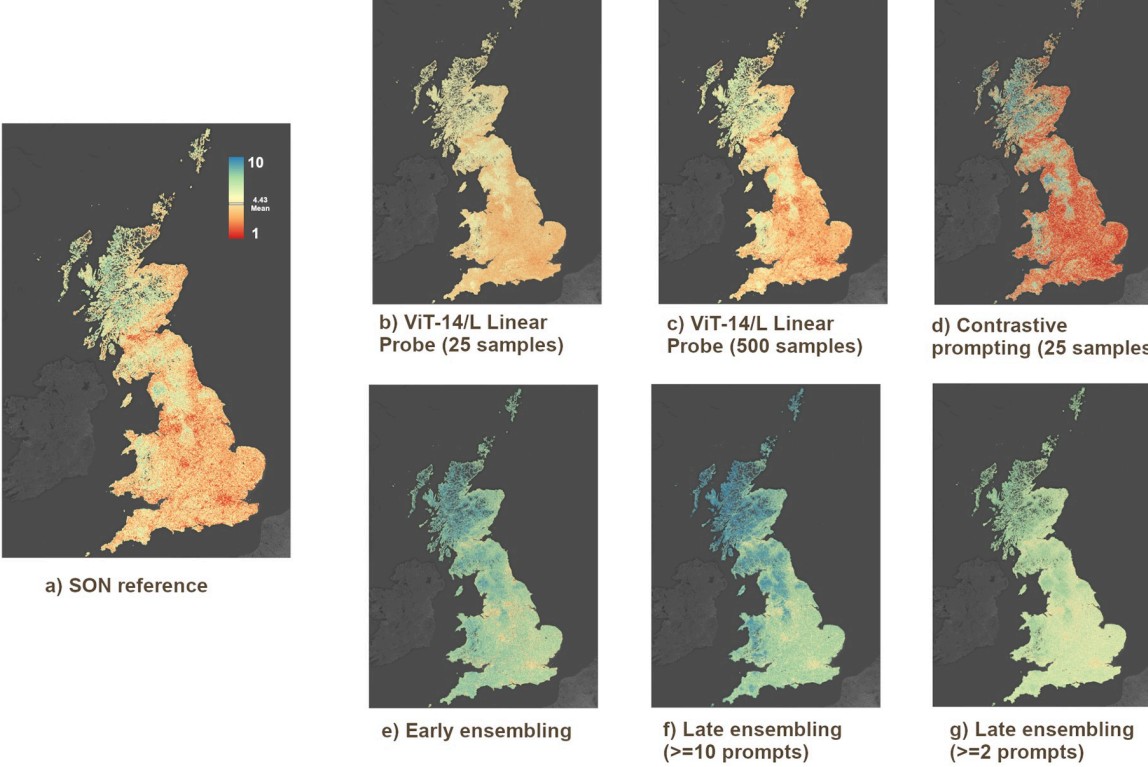

**Fig 8.** Maps of all methods tested in our research compared to the SON reference data (shown in panel a)). The first row (panels b-d-f) showcases methods based on the image ratings of SON, while the second row (panels c-e-g) showcases methods based on the descriptions provided by our volunteers. The predicted scenicness ratings of each method vary greatly, though the main trends between the models are similar, e.g., rugged wilderness being considered more beautiful than man-made areas such as cities.

scenicness distribution. This explains the high RMSE that can be seen in Fig 6, despite both its $R^2$ and Kendall's $\tau$ being competitive compared to the few-show methods. It is therefore likely better to use quantile maps to display scenicness predictions for contrastive prompting. At the country level, all of the presented methods show the most important scenicness patterns, such as low scenicness in cities and high scenicness in elevated areas and shrublands.

We further study the image scenicness ratings obtained by our LPE late ensembling method by analysing the ranking of land cover classes. We sample the level-2 land cover class of the 2018 CORINE inventory [37] at the geolocation of each image within SON. In Fig 9, we show a comparison between the SON image ratings (left) and the ratings calculated by LPE (right) for each land cover class. The plot indicates that there is strong agreement in the ranking of the scenicness of most land cover classes between SON and LPE. One notable exception is that class 1.2 "*Industrial, commercial, and transport units*" is considered to be more scenic by our LPE method when compared to the SON image ratings. However, we do not observe an obvious reason for this increase, such as voters consistently rating industrial elements higher. Among the 14 classes, we observe differences both in the per-class and between-classes distribution of the scenicness predictions. We run a Welch's $t$-test to assess the statistical significance of the scores per class: for most classes, the observed differences are considered significant at the 5% level ($p < 0.05$), with two exceptions: classes 1.2 ('*Industrial, commercial, and transport units*') vs 1.3 ('*Mine, dump and construction sites*') and 2.1 ('*Arable land*') and 2.2

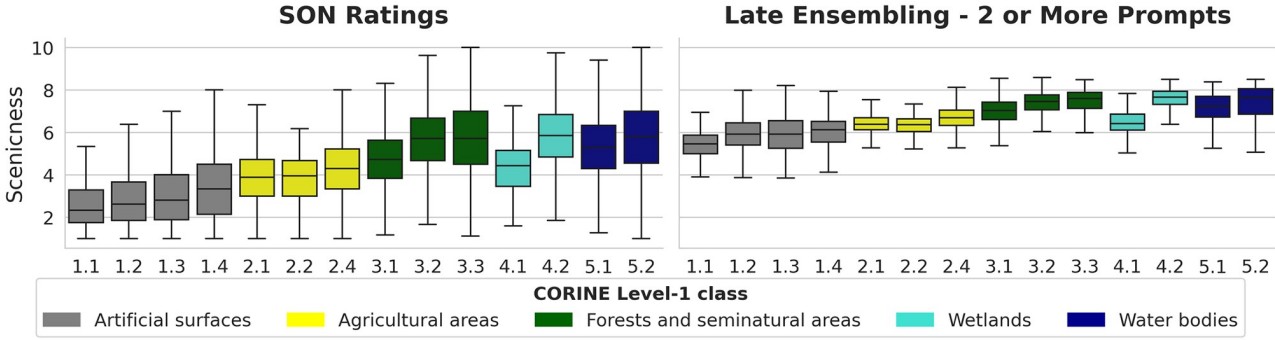

**Fig 9. Comparison of ground truth average image ratings as plotted for each class in SON (left) with the image ratings generated by the LPE method that most closely matched SON in ranking performance (right).** The relative rankings for each land cover class are highly similar, though the LPE mean rating is far higher than in SON. Details about the classes is given in S2 Appendix.

('*Permanent crops*'). In both cases, the classes are highly similar in visual appearance, which might have caused this lack of significant ranking difference. All the classes between groups (i.e. classes with different colors in Fig 9 showed significant differences.

## LPE prompt relevance

A core assumption of prompt-based prediction methods is that prompts should be able to accurately describe an image. With informative prompts, a VLM must be able to recognize the correct image, at least better than random prompts. The LPE method uses the Softmax activations over prompt-image similarities to calculate scenicness scores, which are then weighted according to voters scores. To determine if relevant prompts are indeed chosen for the images, we calculate the Softmax activations of all 138 voter-provided prompts across all images. For each CORINE level-2 land cover class (detailed in S2 Appendix), we then calculate the median confidence for each prompt across all images belonging to that land cover class. In S2 Appendix, we display the most commonly activated prompts for each CORINE land cover class. We observe that the model indeed recognizes relevant prompts for most land cover classes, such as building-related prompts for urban classes (CLC classes in group 1), and rugged terrain prompts for wild areas (CLC classes in group 3). This analysis shows that the voter-provided prompts are indeed suitable, and that pre-trained VLMs are able to select the relevant prompts from the complete set.

## Comparing early and late ensembling

Finally, we studied the effect of the two proposed methodcs for combining prompts into scenicness prediction scores (Fig 5): early ensembling, which relies on finding the most relevant prompt from the complete set of prompts, and late ensembling, which calculates the prompt relevance and scenicness for each voters individually before averaging them across voters. The metrics shown in Table 2 suggest that late ensembling is the best performing ensembling method, since it leads to reduced variance in the scenicness prediction. Considering that models are indeed able to find accurate prompts consistently (see the previous section), the difference in performance may be related to the individual rating given to each prompt by the voters providing it. In late ensembling, this type of variance is averaged out across all voters, while in the early ensembling set-up it may be possible for a single prompt, and therefore a single rating, to dominate the ensemble score. In future research we will explore alternative strategies

for reducing the variance of early ensembling, for instance based on two-stages approaches involving a second group of voters. Finally, the computational complexity of both LPE ensembling methods lends itself well to fast analyses. Consider a dummy dataset of 500'000 pre-computed image embeddings, and 10'000 pre-computed text embeddings from prompts provided by 2'500 voters, running on a single AMD EPYC 7452@2.3GHz CPU core. In this setting, early ensembling takes an average of 4.99 seconds over 10 runs, while late ensembling takes 13.30 seconds on average. As such, even on large datasets, LPE is suitable for use on a smaller computational budget.

## Conclusion

In this paper, we studied the potential for VLMs to reduce the labelling dependency of data-driven scenicness prediction. Firstly, we studied the potential of the image encoder of a CLIP pre-trained transformer model to provide good features for scenicness prediction when fine-tuned with 25 to 500 samples from the ScenicOrNot dataset. Our findings prove that a linear probe fine-tuned on 500 samples is just 5% less accurate than an ImageNet pre-trained model trained on 342 times more samples (Kendall's $\tau$ of 0.625 compared to 0.651).

Secondly, we explored the potential of zero-shot contrastive prompting, which uses a shared prompt context and a synonym and antonym for scenicness (e.g., "beautiful" and "ugly") to find the model's preference for the synonym, which provides a continuous scenicness score. We determined that the choice of prompt context is important for the performance of the method and that a suitable prompt configuration can be discovered using representative samples from the final dataset. The best-performing contrastive prompt configuration tuned to 25 samples outperforms a linear probe when ranking samples. However, it has the tendency to predict extreme values, and as a result, it has a higher standard error.

Lastly, we introduced Landscape Prompt Ensembling (LPE), a new method for the annotation and prediction of landscape scenicness that uses rated textual descriptions of landscapes. Through a small-scale survey, we asked volunteers to provide text descriptions of landscapes they liked or disliked within the United Kingdom, along with a rating for the description. Our ensembling methods then use the confidence of the VLM that a prompt matches an image multiplied by its provided user rating to determine the scenicness contribution of that prompt. We tested if it is better to find the best prompts for a given image out of all prompts provided by all voters (early ensembling), or to calculate the scenicness score for each voter individually before averaging the scenicness scores of all voters (late ensembling). Our results indicate that the latter method provides image scenicness ratings that are more in agreement with ScenicOrNot ratings (Kendall's $\tau$ of 0.485). We also demonstrated that the scenicness ranking of land cover classes of our LPE method is highly similar to the ranking observed for the ScenicOrNot ratings.

Our results show that VLMs have the potential to perform accurate data-driven landscape scenicness assessments on a smaller scale than previously possible and with greater flexibility. We also demonstrated that VLMs can open up new possibilities for landscape scenicness assessments beyond rated image datasets through LPE. We believe that our experiments allow for the prediction of a wider array of landscape qualities than has previously been considered, and we hope that our findings inspire future research experiments with VLMs on other landscape qualities, especially on regional and local scales. Moreover, we want to encourage future work to further investigate the role of vocabulary for aesthetic prediction, and to advance the state of prompting-based methods for landscape quality prediction, for instance through the use of hands-free prompt design methods.

## Supporting information

**S1 Appendix. SigLIP results.**
(PDF)

**S2 Appendix. Most-activated prompts.**
(PDF)

## Author Contributions

**Conceptualization:** Alex Levering, Diego Marcos, Nathan Jacobs, Devis Tuia.

**Formal analysis:** Alex Levering.

**Investigation:** Alex Levering, Diego Marcos.

**Methodology:** Alex Levering, Diego Marcos, Nathan Jacobs.

**Project administration:** Devis Tuia.

**Software:** Alex Levering.

**Supervision:** Diego Marcos, Nathan Jacobs, Devis Tuia.

**Validation:** Alex Levering.

**Visualization:** Alex Levering.

**Writing – original draft:** Alex Levering, Diego Marcos, Nathan Jacobs, Devis Tuia.

**Writing – review & editing:** Alex Levering, Diego Marcos, Nathan Jacobs, Devis Tuia.

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
