## [Decision Letter · Decision Letter 0]

22 Mar 2024

PONE-D-23-40032Prompt-guided and multimodal landscape scenicness assessments with vision-language modelsPLOS ONE

Dear Dr. Tuia,

Thank you for submitting your manuscript to PLOS ONE. After careful consideration, we feel that it has merit but does not fully meet PLOS ONE’s publication criteria as it currently stands. Therefore, we invite you to submit a revised version of the manuscript that addresses the points raised during the review process.

We look forward to receiving your revised manuscript.

Kind regards,

Tinggui Chen

Academic Editor

PLOS ONE

Journal Requirements:

3. You indicated that ethical approval was not necessary for your study. We understand that the framework for ethical oversight requirements for studies of this type may differ depending on the setting and we would appreciate some further clarification regarding your research. Could you please provide further details on why your study is exempt from the need for approval and confirmation from your institutional review board or research ethics committee (e.g., in the form of a letter or email correspondence) that ethics review was not necessary for this study? Please include a copy of the correspondence as an ""Other"" file.

4. Please ensure that you refer to Figure 3 and 5 in your text as, if accepted, production will need this reference to link the reader to the figure.

**Additional Editor Comments:**

I have completed my evaluation of your manuscript. The reviewers recommend reconsideration of your manuscript following major revision. I invite you to resubmit your manuscript after addressing the comments below.

Reviewers' comments:

Reviewer's Responses to Questions

**Comments to the Author**

1. Is the manuscript technically sound, and do the data support the conclusions?

Reviewer #1: Yes

Reviewer #2: Yes

Reviewer #3: Partly

2. Has the statistical analysis been performed appropriately and rigorously? 

Reviewer #1: No

Reviewer #2: Yes

Reviewer #3: Yes

3. Have the authors made all data underlying the findings in their manuscript fully available?

Reviewer #1: Yes

Reviewer #2: Yes

Reviewer #3: Yes

4. Is the manuscript presented in an intelligible fashion and written in standard English?

Reviewer #1: Yes

Reviewer #2: Yes

Reviewer #3: Yes

5. Review Comments to the Author

Reviewer #1: The paper evaluates the capability of Vision Language Models (VLM) for quantifying and mapping landscape scenicness in few-shot and zero-shot settings. An annotation method named Landscape Prompt Ensembling (LPE) is proposed, allowing ensembles of rated text descriptions provided by volunteers to assess landscape scenicness without the need for an image dataset. The results demonstrate promising performance of VLMs in quantifying landscape scenicness and providing assessments beyond the included image datasets through LPE.

The motivation behind this paper is interesting and valuable, and the experiments are well-designed. However, there are several concerns that may help the authors improve the paper's quality:

1. Statistical analysis for Figure 7, Table 2, and Figure 8 is needed for a comprehensive understanding of the performance in different settings. This is crucial to support the conclusions, especially since the size of the testing samples is small.

2. The manuscript seems to imply that late ensembling involves multiple score fusions, including intra-rater scoring with multiple prompts and inter-rater scoring. Clarification on why these differences matter, their computational complexities, and the benefits of the preferred method would be helpful.

3. Comparing different backbones in Figure 6 may not be fair for assessing performance capability in few-shot or zero-shot learning. The model capabilities could have performance gaps due to varying backbones and pretrained weights.

4. The manuscript has several formatting issues. For instance, all Section numbers in the text are missing (e.g., Line 130, 133, 136, 146, 182). Some formulas lack consistency in punctuation (Some of the formulas have commas and periods, but some of them don’t), and all front quotation marks are back quotation marks. Please address these and other format errors in the revised manuscript.

Reviewer #2: 1. The main issue of this manuscript is the novelty. This kind of few-shot setting on CLIP has been widely studied in previous works [13, 28]. Apart from CLIP, recent multimodal large language models (MLLMs), such as GPT-4V [i] and LLaVA [ii], have shown stronger performance. It will be better to add experiments based on them to make this paper more comprehensive.

[i] (arXiv'2309.17421) The Dawn of LMMs: Preliminary Explorations with GPT-4V(ision)

[ii] (NeurIPS'23) Visual Instruction Tuning

2. In L216, it mentions that the CLIP encoder is frozen even for few-shot experiments. I am wondering why it should not be updated. From my understanding, landscape images are mostly similar and require more detailed visual features to extract useful patterns for scenicness prediction. However, CLIP is not pre-trained for this fine-grained scenario, where a fixed CLIP encoder may not extract sufficient features.

3. Currently, all the proposed prompts are annotated by humans. We need to try-and-error and go over to have the best selection among them. Is there any way to find the most adaptive prompt (or even the soft, in the features space, one as [13]) in a hands-free manner? This can make the whole system more practical and ensure further robust performance.

Reviewer #3: The authors evaluate the potential of VLMs in predicting the aesthetic quality of landscapes, and explore the zero-shot prediction potential of contrasting aesthetic concepts and introduce LPE (Landscape Prompt Ensembling).

The paper presents a rich variety of graphical displays. However, it lacks theoretical innovation and lacks theoretical support for the motivation between specific computational steps.

The overall layout of the paper is good, but there are some minor issues, such as multiple instances of "(Section )" appearing incomplete in the Methods section on page five. Please ensure all indices are properly filled.

In terms of experimental design, the authors should incorporate additional methods for similar tasks and provide comparative results with different backbones. This will enhance the comprehensive demonstration of the effectiveness of their proposed methodology.

We look forward to the authors revising and improving the manuscript before resubmission.

6. PLOS authors have the option to publish the peer review history of their article (what does this mean?). If published, this will include your full peer review and any attached files.

Reviewer #1: No

Reviewer #2: No

Reviewer #3: No

---

## [Author Response · Author response to Decision Letter 0]

10 Jun 2024

Please find the reviewer comments in the cover letter submitted with the paper.

---

## [Decision Letter · Decision Letter 1]

1 Jul 2024

Prompt-guided and multimodal landscape scenicness assessments with vision-language models

PONE-D-23-40032R1

Dear Dr. Tuia,

We’re pleased to inform you that your manuscript has been judged scientifically suitable for publication and will be formally accepted for publication once it meets all outstanding technical requirements.

Kind regards,

Tinggui Chen

Academic Editor

PLOS ONE

Additional Editor Comments (optional):

Reviewers' comments:

Reviewer's Responses to Questions

**Comments to the Author**

1. If the authors have adequately addressed your comments raised in a previous round of review and you feel that this manuscript is now acceptable for publication, you may indicate that here to bypass the “Comments to the Author” section, enter your conflict of interest statement in the “Confidential to Editor” section, and submit your "Accept" recommendation.

Reviewer #1: All comments have been addressed

Reviewer #3: All comments have been addressed

2. Is the manuscript technically sound, and do the data support the conclusions?

Reviewer #1: Yes

Reviewer #3: Yes

3. Has the statistical analysis been performed appropriately and rigorously? 

Reviewer #1: Yes

Reviewer #3: Yes

4. Have the authors made all data underlying the findings in their manuscript fully available?

Reviewer #1: Yes

Reviewer #3: Yes

5. Is the manuscript presented in an intelligible fashion and written in standard English?

Reviewer #1: Yes

Reviewer #3: Yes

6. Review Comments to the Author

Reviewer #1: I appreciate the additional experiments and further clarification in the revised manuscript. All comments from the previous review have been addressed.

Reviewer #3: Thank the author for adopting the review suggestions. The revisions in this version are much better, and I look forward to the final publication.

7. PLOS authors have the option to publish the peer review history of their article (what does this mean?). If published, this will include your full peer review and any attached files.

Reviewer #1: No

Reviewer #3: No

---

## [Editor Report · Acceptance letter]

9 Aug 2024

PONE-D-23-40032R1 

PLOS ONE

Dear Dr. Tuia, 

I'm pleased to inform you that your manuscript has been deemed suitable for publication in PLOS ONE. Congratulations! Your manuscript is now being handed over to our production team.

Kind regards, 

on behalf of

Dr. Tinggui Chen 

Academic Editor

PLOS ONE